# SARS-CoV-2 suppresses anticoagulant and fibrinolytic gene expression in the lung

Alan E Mast[1†], Alisa S Wolberg[2†], David Gailani[3†], Michael R Garvin[4], Christiane Alvarez[4], J Izaak Miller[4], Bruce Aronow[5,6,7], Daniel Jacobson[4,5,8]*

[1]Versiti Blood Research Institute, Department of Cell Biology Neurobiology and Anatomy Medical College of Wisconsin, Milwaukee, United States; [2]Department of Pathology and Laboratory Medicine and UNC Blood Research Center, Chapel Hill, United States; [3]Department of Pathology, Microbiology and Immunology, Vanderbilt University Medical Center, Nashville, United States; [4]Oak Ridge National Laboratory, Biosciences Division, Oak Ridge, United States; [5]University of Tennessee Knoxville, The Bredesen Center for Interdisciplinary Research and Graduate Education, Knoxville, United States; [6]Biomedical Informatics, Cincinnati Children's Hospital Research Foundation, Cincinnati, United States; [7]University of Cincinnati, Cincinnati, United States; [8]University of Tennessee Knoxville, Department of Psychology, Knoxville, United States

**Abstract** Extensive fibrin deposition in the lungs and altered levels of circulating blood coagulation proteins in COVID-19 patients imply local derangement of pathways that limit fibrin formation and/or promote its clearance. We examined transcriptional profiles of bronchoalveolar lavage fluid (BALF) samples to identify molecular mechanisms underlying these coagulopathies. mRNA levels for regulators of the kallikrein–kinin (C1-inhibitor), coagulation (thrombomodulin, endothelial protein C receptor), and fibrinolytic (urokinase and urokinase receptor) pathways were significantly reduced in COVID-19 patients. While transcripts for several coagulation proteins were increased, those encoding tissue factor, the protein that initiates coagulation and whose expression is frequently increased in inflammatory disorders, were not increased in BALF from COVID-19 patients. Our analysis implicates enhanced propagation of coagulation and decreased fibrinolysis as drivers of the coagulopathy in the lungs of COVID-19 patients.

*For correspondence:
jacobsonda@ornl.gov

[†]These authors contributed equally to this work

## Introduction

The bradykinin storm model for COVID-19 pathogenesis was recently developed from our analyses of gene expression, clinical, autopsy, pathology, and ChIP-Seq data (*Garvin et al., 2020*). Several clinical studies by other groups have demonstrated positive results from therapeutic interventions predicted by our model (*Entrenas Castillo et al., 2020*; *van de Veerdonk et al., 2020*; *WHO Rapid Evidence Appraisal for COVID-19 Therapies (REACT) Working Group et al., 2020*). Here, we extend this model to include the concurrent dysregulation of the coagulation and fibrinolytic pathways.

SARS-CoV-2 displays considerable tissue tropism (*Adachi et al., 2020*; *Barton et al., 2020*; *Su et al., 2020*; *Xu et al., 2020*). The lungs are often affected with disease that can range from mild pneumonia, to severe dyspnea and hypoxia, to critical respiratory failure, shock, and multiorgan failure. The time course for the development of severe disease is typically 8–12 days, but some patients rapidly deteriorate about 7 days following development of symptoms. Lung tissue from people who died from COVID-19 has pauci-inflammatory septal capillary injury and luminal and mural fibrin deposition in alveolar septal capillaries (*Lax et al., 2020*; *Magro et al., 2020*), thrombi in small- and

medium-sized arteries (*Lax et al., 2020*; *Magro et al., 2020*), and fibrinous thrombi in small pulmonary arterioles with evidence of tumefaction of the endothelium (*Dolhnikoff et al., 2020*; *Fox et al., 2020*; *Geerdes-Fenge et al., 2020*; *Lax et al., 2020*). A report that administering tissue plasminogen activator (tPA) to dissolve fibrin transiently alleviates respiratory distress in COVID-19 patients supports the premise that fibrin deposition contributes to the respiratory failure (*Wang et al., 2020*). Thus, it appears that COVID-19 hypoxemia stems at least partially from fibrin deposits surrounding alveoli that restrict oxygen transfer and microvascular thrombi that cause ventilation–perfusion defects.

Abnormal levels of circulating coagulation proteins are present in patients with COVID-19. Changes include increased levels of coagulation proteins associated with the acute phase response (e.g., fibrinogen and factor VIII) and endothelial activation (e.g., von Willebrand factor), as well as elevated biomarkers of coagulation activation (e.g., D-dimer) (*Chen et al., 2020*; *Gattinoni et al., 2020*; *Helms et al., 2020*; *Middeldorp et al., 2020*; *Panigada et al., 2020*; *Tang et al., 2020*; *Zhou et al., 2020a*; *Helms et al., 2020*; *Middeldorp et al., 2020*; *Panigada et al., 2020*). Accordingly, thrombotic events have been detected in up to 30% of COVID-19 patients (*Helms et al., 2020*; *Middeldorp et al., 2020*; *Panigada et al., 2020*) including large vessel occlusions such as deep vein thrombosis, pulmonary embolism, and ischemic stroke, as well as microvascular thrombosis and extravascular fibrin deposition in a variety of tissues including lung and skin (*Helms et al., 2020*; *Middeldorp et al., 2020*; *Panigada et al., 2020*). Inappropriate fibrin deposition and thrombosis are thought to stem from the interaction of systemic (blood) changes with tissue-specific dysfunction. However, mechanisms that contribute to prevalence of thrombi and fibrin in the pulmonary vasculature and extravascular space, thought to be a major cause of morbidity and mortality in COVID-19, have remained elusive.

The goal of the present study was to identify changes in the lungs of patients with severe COVID-19 that could contribute to local derangement of hemostatic mechanisms. Bronchoalveolar lavage fluid (BALF) contains lung parenchymal, epithelial, and alveolar cells, as well as immune cells that infiltrate epithelial and luminal spaces. RNA sequencing of BALF provides a snapshot of the transcriptome at the interface where capillary gas exchange occurs and has been used to characterize lung function in many diseases (*Gu et al., 2019*; *Jakieła et al., 2018*; *Kahn et al., 2015*; *Sengupta et al., 2019*; *Sun et al., 2019*; *Weigt et al., 2019*; *Zhang et al., 2019*). Here, we compared the transcriptional signatures of BALF from patients with COVID-19 and uninfected controls. Our analyses focused on transcripts for proteins that function in coagulation, fibrinolysis, and kinin formation in the lung to identify dysregulated mechanisms that may contribute to COVID-19 pathophysiology.

## Results and discussion

### Changes in transcripts encoding proteins in the kallikrein–kinin/bradykinin system

#### Overview of pathways examined

The plasma kallikrein–kinin system is comprised of the protease precursors factor XII (FXII, encoded by F12) and prekallikrein (encoded by *KLKB1*) and the cofactor high-molecular-weight kininogen (HK, encoded by *KNG1*) (*Schmaier, 2016*; *Schmaier et al., 2019*). In healthy individuals, FXII and prekallikrein undergo reciprocal activation to the proteases FXIIa and kallikrein (*Revenko et al., 2011*; *Schmaier, 2016*). Kallikrein cleaves HK to liberate bradykinin, which contributes to setting vascular tone and permeability by interacting with bradykinin receptors (encoded by *BDKRB1* and *BDKRB2*) (*Marceau et al., 2020*). This process is regulated by C1-Inhibitor (encoded by *SERPING1*) (*Marceau et al., 2020*). Congenital C1-Inhibitor deficiency causes hereditary angioedema, which is characterized by bouts of bradykinin-induced soft tissue swelling (*Busse and Christiansen, 2020*; *Cicardi and Zuraw, 2018*). C1-Inhibitor in blood is primarily of hepatocyte origin, although it also is produced by other cell types including vascular endothelium (*Prada et al., 1998*).

#### Study findings

Transcripts encoding C1-Inhibitor were decreased 80-fold in BALF from COVID-19 patients (*Table 1*), raising the possibilities that contact activation-initiated thrombin generation is locally dysregulated

**Table 1.** Differentially expressed coagulation genes.

| Gene | Protein product | Mean COVID-19 | Mean control | Fold change | Log2FC | FDR |
|---|---|---|---|---|---|---|
| A2M | α2-Macroglubulin | 3.9 | 177.6 | −43.5 | −5.4 | 5.2E-15 |
| BDKRB1 | Bradykinin receptor B1 | 3.3 | 0.0 | 258.9 | 8.0 | 4.3E-91 |
| BDKRB2 | Bradykinin receptor B2 | 8.9 | 0.2 | 49.1 | 5.6 | 1.9E-40 |
| F13A1 | Factor XIII-A subunit | 2.5 | 9.2 | −3.6 | −1.8 | 2.6E-05 |
| F13B | Factor XIII-B subunit | 0.6 | 0.0 | 117.8 | 6.9 | 2.7E-31 |
| F12 | Factor XII | 0.5 | 3.2 | −4.4 | −2.1 | 5.7E-10 |
| F11 | Factor XI | 6.6 | 0.1 | 81.2 | 6.3 | 7.9E-79 |
| F10 | Factor X | 3.2 | 0.0 | 169.9 | 7.4 | 7.9E-140 |
| F9 | Factor IX | 0.9 | 0.0 | 189.6 | 7.6 | 2.2E-39 |
| F8 | Factor VIII | 3.9 | 15.8 | −4.6 | −2.2 | 1.6E-40 |
| F7 | Factor VII | 10.1 | 0.0 | 363.5 | 8.5 | 2.9E-73 |
| F5 | Factor V | 9.7 | 6.8 | 1.4 | 0.5 | 0.13 |
| F3 | Tissue Factor | 4.2 | 3.8 | -1.0 | -0.01 | 1 |
| F2 | Prothrombin | 0.9 | 0.0 | 188.8 | 7.6 | 7.8E-67 |
| FGA | Fibrinogen Aα chain | 3.4 | 0.0 | 322.5 | 8.3 | 3.4E-206 |
| FGB | Fibrinogen Bβ chain | 3.7 | 0.0 | 303.9 | 8.2 | 2.4E-151 |
| FGG | Fibrinogen γ chain | 0.7 | 0.0 | 85.4 | 6.4 | 4.8E-17 |
| KLKB1 | Kallikrein B1 | 1.3 | 1.8 | −1.4 | −0.5 | 2.1E-01 |
| KNG1 | Kininogen | 3.3 | 0.0 | 190.7 | 7.6 | 4.2E-163 |
| PLAT | Tissue plasminogen activator | 1.8 | 0.4 | 5.3 | 2.4 | 7.1E-24 |
| PLAU | Urokinase | 3.8 | 158.3 | −37.1 | −5.2 | 8.2E-172 |
| PLAUR | Urokinase receptor | 6.2 | 313.3 | −42.1 | −5.4 | 6.4E-286 |
| PLG | Plasminogen | 2.2 | 0.0 | 75.3 | 6.2 | 7.3E-38 |
| PROC | Protein C | 2.0 | 0.0 | 226.5 | 7.8 | 4.1E-158 |
| PROCR | Endothelial protein C receptor | 1.7 | 57.9 | −33.8 | −5.1 | 1.9E-50 |
| PROS1 | Protein S | 3.2 | 195.3 | −54.2 | −5.8 | 1.1E-174 |
| SERPINA5 | Protein C Inhibitor | 2.7 | 0.0 | 786.8 | 9.6 | 5.4E-145 |
| SERPINB2 | Plasminogen activator inhibitor-2 | 1.0 | 0.5 | 2.3 | 1.2 | 2.1E-01 |
| SERPINC1 | Antithrombin | 1.9 | 0.1 | 13.6 | 3.8 | 3.1E-31 |
| SERPIND1 | Heparin cofactor II | 3.0 | 0.0 | 94.0 | 6.6 | 2.2E-72 |
| SERPINE1 | Plasminogen activator inhibitor-1 | 2.7 | 5.3 | −1.8 | −0.9 | 8.8E-02 |
| SERPINF2 | α2-antiplasmin | 5.4 | 8.2 | −1.4 | −0.5 | 1.1E-02 |
| SERPING1 | C-1 inhibitor | 23.3 | 1923.2 | −80.1 | −6.3 | 5.4E-33 |
| TFPI | Tissue factor pathway inhibitor | 3.4 | 0.4 | 7.7 | 2.9 | 8.7E-19 |
| THBD | Thrombomodulin | 9.6 | 224.0 | −22.2 | −4.5 | 3.2E-48 |
| VWF | Von Willebrand factor | 14.5 | 7.7 | 2.0 | 1.0 | 2.0E-05 |

and control of bradykinin production is compromised. Furthermore, the angiotensin-converting enzyme (ACE) that degrades bradykinin (*Cicardi and Zuraw, 2018*; *Davin et al., 2019*) was downregulated eightfold in COVID-19 BALF.

## Changes in transcripts encoding extrinsic pathway proteins that initiate thrombin generation

### Overview of pathways examined

The integral cytoplasmic membrane protein tissue factor (TF, encoded by *F3*) is the primary initiator of thrombin generation in vivo. In many inflammatory conditions, cellular TF expression is upregulated (*Edwards et al., 1979*), providing a mechanistic explanation for hypercoagulable states associated with a variety of bacterial and viral infections (*Grover and Mackman, 2018*; *Levi and van der Poll, 2017*). Tissue factor pathway inhibitor (TFPI, encoded by *TFPI*) is a Kunitz-type protease inhibitor produced by vascular endothelium (*Bajaj et al., 1990*) and is the major inhibitor of the factor VIIa (FVIIa)/TF complex (*Wood et al., 2014*).

### Study findings

Surprisingly, transcripts encoding TF were similar in COVID-19 and control BALF samples (*Figure 1*; *Table 1*), whereas transcripts for TFPI were increased eightfold in COVID-19 BALF. On balance, these data indicate that pulmonary fibrin deposition does not stem from enhanced local TF production and that counterintuitively, COVID-19 may dampen TF-dependent mechanisms in the lungs.

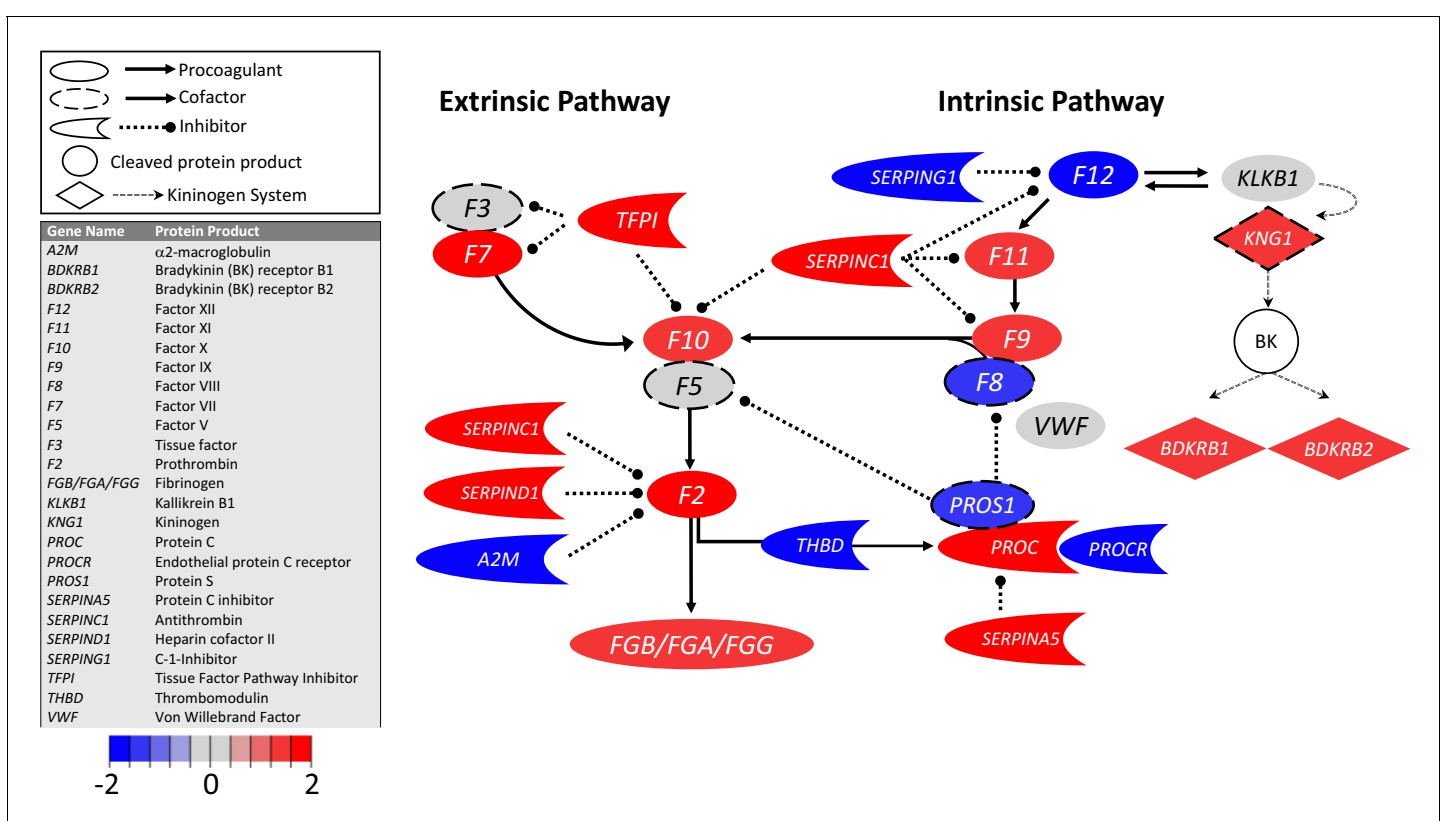

**Figure 1.** Transcriptional changes in lung induced by COVID-19 infection did not alter *F3*, but decreased aPC anticoagulant capacity, suggesting decreased inhibition of the propagation phase of coagulation. Figure shows differential gene expression (log$_2$ fold change) of coagulation pathway transcripts in BALF of COVID-19 patients; the image illustrates mechanistic relationships of the protein products of the identified transcripts during coagulation. Shading indicates relative expression in COVID-19 patients compared to controls: increased (red) or decreased (blue). There was minimal change in *F3* (encoding tissue factor ) and increased *TFPI* (encoding the major inhibitor of tissue factor activity). There was decreased *THBD, PROCR*, and *PROS1* (encoding proteins that enhance anticoagulant activity) and increased *SERPINA5* (encoding protein C inhibitor). There was also decreased *SERPING1* (encoding C1-Inhibitor). Other transcripts showing changes (e.g., F11, F10, F7, F2) encode proteins typically produced in the liver; local expression of these proteins is unclear.

## Changes in transcripts encoding intrinsic pathway proteins that initiate thrombin generation

### Overview of pathways examined

Increased thrombin generation can also be initiated by the plasma protease FXIIa (*Maas and Renné, 2018*; *Stavrou, 2018*). While FXIIa does not contribute to hemostasis, it is implicated in thrombo-inflammatory conditions. FXIIa converts factor XI (FXI, encoded by *F11*) to the protease FXIa (*Ivanov et al., 2017*; *Long et al., 2016*; *Schmaier, 2016*; *Wang et al., 2019*), a potent activator of the coagulation protease factor IX (FIX, encoded by *F9*) (*Mohammed et al., 2018*).

### Study findings

Transcripts encoding FXII were slightly decreased (−4-fold) in COVID-19 and control BALF samples (*Figure 1*; *Table 1*), but the expression of the C1 inhibitor (encoded by *SERPING1*), which normally inhibits the FXII protein activity, is downregulated 80-fold. While it is not known if pulmonary tissues normally produce FXI or FIX, transcripts for FXI (81-fold) and FIX (190-fold) were both increased in BALF from COVID-19 patients. Numerous studies have reported correlations between plasma FIX and FXI levels and risk for venous thromboembolism and ischemic stroke (*Ammollo et al., 2014*; *Folsom et al., 2015*; *Mohammed et al., 2018*; *Preis et al., 2017*; *Salomon et al., 2011*, *Salomon et al., 2008*; *Siegerink et al., 2010*; *Suri et al., 2010*). Perhaps local increases in FIX or FXI expression enhance thrombin generation and promote thrombus formation.

## Changes in transcripts encoding anticoagulant proteins

### Overview of pathways examined

Healthy lung endothelium has membrane-associated anticoagulant proteins that inhibit different points in the thrombin generation pathway. These include TFPI, which inhibits the initiation of coagulation as described above, and thrombomodulin (encoded by *THBD*) and endothelial protein C receptor (EPCR, encoded by *PROCR*), which coordinate to inhibit the propagation of coagulation. Thrombomodulin captures thrombin and converts it from a procoagulant enzyme to an activator of protein C (encoded by *PROC*) in a process enhanced by binding of PC to EPCR. Activated PC with its cofactor protein S (encoded by *PROS1*) downregulates thrombin generation by inactivating factors VIIIa and Va and produces a cytoprotective/anti-inflammatory effect when bound to EPCR through cleavage of protease activated receptor-1 on endothelial cells (*Ito et al., 2019*; *Riewald et al., 2002*).

### Study findings

Transcripts encoding thrombomodulin (−22-fold) and EPCR (−33-fold) were each reduced in BALF from COVID-19 patients compared with controls (*Figure 1*; *Table 1*), suggesting SARS-CoV-2 is associated with reduced expression of these proteins on vascular endothelium. We also observed a marked increase in protein C transcripts in BALF from COVID-19 patients (227-fold), but a significant reduction in PS transcripts (−54-fold). Since protein C and protein S in blood are primarily synthesized by hepatocytes, the importance of local synthesis of these proteins is not known. However, expression patterns in BALF are consistent with the observation that the plasma concentration of protein C is moderately elevated, and protein S moderately reduced, in COVID-19 patients (*Panigada et al., 2020*). In COVID-19 BALF, there was also a substantial increase (786-fold) in mRNA for protein C inhibitor (encoded by *SERPINA5*), a serpin regulator of activated protein C. Taken as a whole, the transcript pattern in COVID-19 BALF suggests a diminished capacity of the endothelial anticoagulant system to downregulate local thrombin generation.

## Changes in transcripts that encode fibrinogen and proteins in the fibrinolytic pathway

### Overview of pathways examined

The fibrinolytic system is responsible for enzymatic degradation of fibrin. Plasminogen activators convert plasminogen to plasmin, which cleaves fibrin and generates fibrin degradation products, including the circulating biomarker D-dimer. Cells within the lung express plasminogen activators (e. g., [tPA, encoded by *PLAT*], urokinase plasminogen activator [uPA, encoded by *PLAU*], and the uPA

receptor [uPAR, encoded by *PLAUR*]) that prevent fibrin accumulation in small airways and the alveolar compartment and that maintain blood vessel patency (*Shetty et al., 2008*). uPA and uPAR are expressed by lung epithelial cells, alveolar macrophages, and fibroblasts, and reduced expression of these proteins contributes to acute lung injury (*Shetty et al., 2008*). The plasminogen activators are inhibited by plasminogen activator inhibitor-1 (PAI-1, encoded by *SERPINE1*). Plasma fibrinogen, a dimer of trimers ($A\alpha_2B\beta_2\gamma_2$) encoded by three genes, *FGA*, *FGB*, and *FGG*, is synthesized primarily by hepatocytes (*Pieters and Wolberg, 2019*); however, synthesis has been reported in stimulated cultured lung alveolar epithelial cells and in alveolar epithelium in an animal model of bacterial pneumonia (*Guadiz et al., 1997*; *Simpson-Haidaris et al., 1998*).

## Study findings

Transcripts encoding the fibrinogen chains were increased in COVID-19 BALF (*Table 1*, *Figure 2*; ). Our findings suggest that local fibrinogen expression in the lungs of COVID-19 patients, like fibrinogen expression by hepatocytes, is upregulated during the acute phase response and may provide additional substrate for local thrombin-mediated fibrin production. Our analysis uncovered evidence of reduced expression of transcripts encoding uPA (−37-fold) and uPAR (−42-fold) (*Table 1*, *Figure 2*) in BALF from COVID-19 patients. Expression of transcripts encoding tPA was very low in BALF from healthy individuals. Transcripts encoding tPA were elevated (fivefold) in COVID-19 BALF, but absolute expression was low compared to uPA and uPAR transcripts in uninfected controls (*Table 1*, *Figure 2*). Similarly, expression of transcript for plasminogen (encoded by *PLG*) was elevated in COVID-19 BALF but remained low compared to other transcripts. The combination of increased fibrinogen expression and reduced production of uPA and uPAR indicates a loss of local fibrinolytic capacity, consistent with a lung environment permissive of the accumulation of fibrin deposits within pulmonary vessels and the alveolar space (*Lax et al., 2020*; *Magro et al., 2020*).

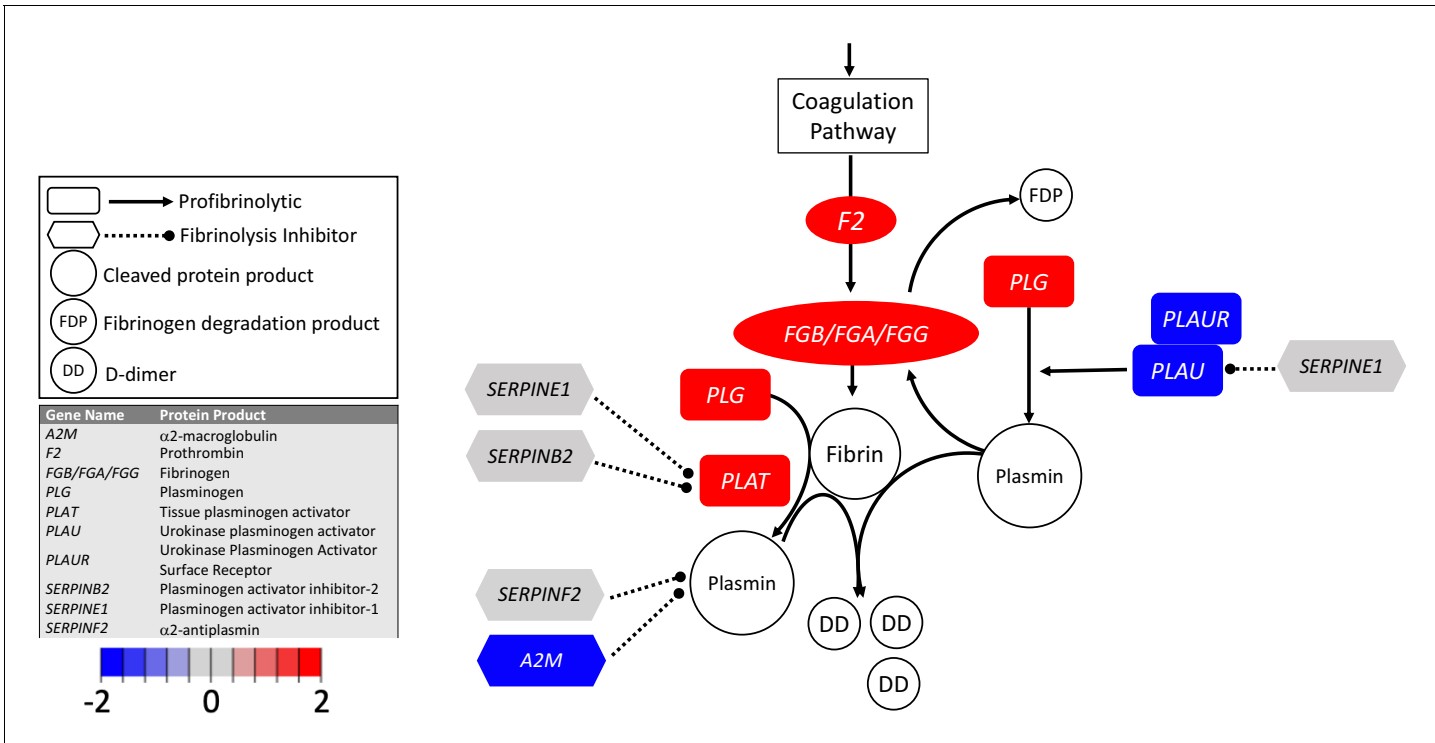

**Figure 2.** Transcriptional changes in lung induced by COVID-19 infection decreased *PLAU* and *PLAUR*, suggesting diminished fibrinolytic activity. Figure shows differential gene expression (log₂ fold change) of fibrinolytic pathway transcripts in BALF of COVID-19 patients; the image illustrates mechanistic relationships of the protein products of the identified transcripts during fibrinolysis. Shading indicates relative expression in COVID-19 patients compared to controls: increased (red) or decreased (blue). There was a moderate increase in *PLAT* (encoding tPA). There was also enhanced expression of *FGB*, *FGA*, and *FGG* (encoding fibrinogen chains) and decreased expression of *PLAU* and *PLAUR* (encoding uPA and uPAR, respectively). Other transcripts showing changes (e.g., *F2*, *PLG*) encode proteins typically produced in the liver; local expression of these proteins is unclear.

## Conclusions

We detected pronounced changes in mRNA levels encoding proteins involved in regulation of coagulation, fibrinolysis, and inflammation in the lungs of COVID-19 patients. These changes include reductions in transcripts for proteins known to be expressed in normal pulmonary tissue (thrombomodulin, EPCR, uPA, uPAR) that would compromise the functions of anticoagulant and fibrinolytic pathways. In concert with enhanced production of bradykinin and vascular permeability, these changes are likely to create an environment in which plasma proteins are exposed to the extravascular space, resulting in increased fibrin production and reduced fibrin degradation, enabling the fibrin deposits in pulmonary vessels and alveolar spaces observed in autopsies of COVID-19 patients (*Dolhnikoff et al., 2020*; *Fox et al., 2020*; *Geerdes-Fenge et al., 2020*; *Lax et al., 2020*), and contributing to the virulence of the virus.

An unexpected finding was the lack of change in transcripts for *F3*, which encodes TF, the integral membrane protein responsible for triggering thrombin generation during hemostasis. TF expression is enhanced in a wide range of thrombo-inflammatory disorders, and its over-expression is thought to be important for driving consumptive coagulation. The apparent lack of increase in *F3*, coupled with an apparent increase in *TFPI* transcripts, raises the possibility that pathways unrelated to the FVIIa/TF complex may play a significant role in COVID-19-associated thrombosis. In this regard, thrombomodulin and EPCR transcripts were substantially decreased, suggesting reduced capacity of the lung endothelium to downregulate the propagation of thrombin generation. In addition, changes in regulation or activation of the plasma kallikrein–kinin system and contact activation may be important. This is consistent with our hypothesis that COVID-19 triggers a bradykinin storm (*Garvin et al., 2020*), the net effect of increased bradykinin production, reduced BK degradation, and increased local expression of bradykinin receptors, is expected to enhance vasodilation and vascular permeability, exposing blood components to extravascular proteins that promote fibrin formation and contribute to tissue damage.

Fibrin deposits and thrombi have been observed in organs other than the lungs in COVID-19 patients, including kidney (*Su et al., 2020*) and spleen (*Mestres et al., 2020*). It is tempting to speculate that transcriptional changes observed in BALF may occur in these other tissues. Currently, it is not clear whether fibrin deposition in specific organs indicates localized infection of those organs, perhaps stemming from differential distribution of the SARS-CoV-2 receptor ACE2, or reflects organ susceptibility to systemic changes induced by the infection. Analysis of transcriptional and proteomic changes in different organs in concert with information on virus entry into parenchymal cells may help explain the pattern of organ-specific thrombosis during SARS-CoV-2 infection.

Our study has limitations. Although we detected changes in multiple gene transcripts, we did not identify specific cell types for each transcript, which may include lung parenchymal cells, leukocytes, megakaryocytes (*Fox et al., 2020*), and other cells present in native and inflammatory states that undergo transcriptional changes in response to infection. Single-cell analyses will be required to better understand these changes during SARS-CoV-2 infection. As with all transcriptional data, we could not determine whether changes in mRNA manifest as changes in protein expression. While histologic analyses clearly demonstrate fibrin deposition in the lung, further studies using agnostic and targeted proteomics, as well as biochemical analyses will be needed to confirm causation. We focused on changes in coagulation and fibrinolytic pathways in the present study; however, the pathologic processes observed in the lungs of COVID-19 patients are likely to be influenced by changes in other pathways, such as those involving complement. Crosstalk between coagulation, fibrinolytic, and complement pathways is well documented (*Ewald and Eisenberg, 1995*; *Foley et al., 2016*; *Fox et al., 2020*; *Kleniewski and Donaldson, 1987*). Despite these limitations, it is instructive to consider how the changes in coagulation and fibrinolytic gene expression documented in our study of BALF may represent common events occurring in multiple vascular beds in COVID-19, as identification of a common dysfunction may reveal a targetable mechanism to prevent thrombosis in these patients.

## Materials and methods

### Study participants

The nine BALF samples from 5 patients used in this study were collected from patients in Wuhan, China. All patients displayed pneumonia and other severe symptoms upon admission to the hospital in Wuhan, China, in late December 2019 and were therefore admitted to the ICU (*Table 2*). They were originally used for RNA sequencing to identify the etiological agent for COVID-19 and to determine the genomic sequence of SARS-CoV-2. Human mRNA sequences from these samples were used to develop our bradykinin storm model (*Garvin et al., 2020*), but have not previously been analyzed to understand localized coagulopathy in the lungs of patients with COVID-19 (*Zhou et al., 2020b*). We compared mRNA transcripts for proteins involved in coagulation, fibrinolysis, and kinin formation in BALF from these samples with those from 40 controls from a study on the contribution of obesity to disease severity in asthmatics (*Michalovich et al., 2019*). The 40 control individuals included smokers and non-smokers, asthmatics and non-asthmatics, as well as obese and non-obese individuals. Given this diverse set of morbidities and the resulting variance in gene expression, they represent an informative control for patients with COVID-19 as statistically significant changes detected due to COVID-19 are not just representing subtle shifts in variance (*Table 1*). Expression datasets are available from the NCBI Sequence Read Archive under the accession numbers PRJNA605983 (severe acute respiratory syndrome coronavirus 2 raw sequence reads) and PRJNA434133 (Microbiome and Inflammatory Interactions in Obese and Severe Asthmatic Adults).

### Gene expression analysis

The CLC Genomics Workbench (20.0.3) was used to trim FASTQ files downloaded from the NCBI Sequence Read Archive to remove any adapter sequences or other artifacts from processing. The RNA reads were mapped to the GRCh38_latest_rna.fna version of the human transcriptome. Mapping parameters included a cost penalty of two for mismatches and three for small insertions or deletions (INDEL), and similarity and length fraction were both set to 0.95. The resulting transcript profiles were manually inspected to account for expression artifacts, such as reads mapping solely to repetitive elements such as the *Alu* transposable element. Transcripts whose counts came solely from (or were dominated by) reads at repetitive elements were removed from the analysis. Where transcripts per million (TPM) values were zero (transcripts were not detected), we used the lowest TPM across samples to two decimal places. If all samples in a group had no detected transcripts, we used the lowest TPM to two decimal places from the entire set of genes queried. The edgeR package (*McCarthy et al., 2012*; *Robinson et al., 2010*) was used to identify genes that were differentially expressed in COVID-19 patient samples compared to controls. Count data were scaled to normalize for library size and normalization factors determined as instructed in edgeR documentation. Dispersion was estimated, and the read counts for each gene were fit with a negative binomial

**Table 2.** Clinical data for patients from which BALF was extracted and analyzed for this study.

| GISAID accession | Isolate | NCBI accession | SRA accessions | Swab date | Age | Sex | Patient no | Date onset | Symptoms_admission | Status 1/13/20 | Diagnosis history |
|---|---|---|---|---|---|---|---|---|---|---|---|
| EPI_ISL_402127 | WIV02 | MN996527 | SRR11092058, SRR11092063 | 12/30/19 | 32 | Male | ICU-04 | 12/19/19 | Fever, cough, dyspnea | Fever, intermitttent cough | Negative |
| EPI_ISL_402124 | WIV04 | MN996528 | SRR11092057, SRR11092062 | 12/30/19 | 49 | Female | ICU-06 | 12/27/19 | Fever (37.9C), palpitation | Fever, malaise, cough | Coronavirus (nt) |
| EPI_ISL_402128 | WIV05 | MN996529 | SRR11092061 | 12/30/19 | 52 | Female | ICU-08 | 12/22/19 | Fever (38C), expectoration, malaise, dyspnea | Recovered, disharged | *Streptococcus pneumoniae* (nt) |
| EPI_ISL_402129 | WIV06 | MN996530 | SRR11092056, SRR11092060 | 12/30/19 | 40 | Male | ICU-09 | 12/28/20 | Fever (38C), expectoration | Fever (38C), expectoration, dizziness | Negative |
| EPI_ISL_402130 | WIV07 | MN996531 | SRR11092059, SRR11092064 | 12/30/19 | 56 | Male | ICU-10 | 12/20/19 | Fever, dyspnea, chest tightness | Fever, malaise, cough, dyspnea | Negative |

model. Each gene was tested for differential expression. The Benjamini–Hochberg method was used to determine the false discovery rate.

## Acknowledgements

We would like to acknowledge funding from the Oak Ridge National Laboratory, Laboratory Directed Research and Development Fund, LOIS:10074 (which supported the systems biology work), and the National Institutes of Health, U24 HL148865: the LungMap Consortium (which supported tissue and cell-based expression conceptualization). Funding was also provided by the National Institutes of Health grants HL068835 (AM), HL143403 (AW), HL126974 (AW), 3RF1AG053303-01S2 (DJ), and HL140025 (DG) to support analyses and interpretation of the coagulation and fibrinolytic pathways. This research used resources of the Oak Ridge Leadership Computing Facility, which is a DOE Office of Science User Facility supported under Contract DE-AC05-00OR22725. This research used resources of the Compute and Data Environment for Science (CADES) at the Oak Ridge National Laboratory.

## Additional information

### Competing interests

Alan E Mast: receives research funding from Novo Nordisk and has received honoraria for serving on Novo Nordisk advisory boards. Alisa S Wolberg: receives research funding from Takeda and Bristol Myers Squibb. David Gailani: receives research funding from Bayer and has received honoraria for serving on Anthos, Bristol-Myers Squibb, Ionis and Janssen advisory boards. The other authors declare that no competing interests exist.

### Funding

| Funder | Grant reference number | Author |
|---|---|---|
| Oak Ridge National Laboratory | LOIS:10074 | Michael R Garvin<br>J Izaak Miller<br>Daniel Jacobson |
| National Institutes of Health | U24 HL148865 | Bruce Aronow |
| National Institutes of Health | HL068835 | Alan E Mast |
| National Institutes of Health | HL143403 | Alisa S Wolberg |
| National Institutes of Health | HL126974 | Alisa S Wolberg |
| National Institutes of Health | HL140025 | David Gailani |
| National Institute on Aging | 3RF1AG053303-01S2 | Daniel Jacobson<br>Michael R Garvin |

The funders had no role in study design, data collection and interpretation, or the decision to submit the work for publication.

### Author contributions

Alan E Mast, Alisa S Wolberg, David Gailani, Bruce Aronow, Conceptualization, Formal analysis, Investigation, Visualization, Writing - original draft, Writing - review and editing; Michael R Garvin, Conceptualization, Data curation, Formal analysis, Visualization, Writing - original draft, Writing - review and editing; Christiane Alvarez, Visualization; J Izaak Miller, Formal analysis, Investigation, Methodology; Daniel Jacobson, Conceptualization, Supervision, Funding acquisition, Investigation, Visualization, Methodology, Writing - original draft, Project administration, Writing - review and editing

### Author ORCIDs

Alan E Mast  https://orcid.org/0000-0003-3740-0318
Alisa S Wolberg  http://orcid.org/0000-0002-2845-2303

Michael R Garvin [ID] https://orcid.org/0000-0002-2204-7569
Daniel Jacobson [ID] https://orcid.org/0000-0002-9822-8251

**Decision letter and Author response**
Decision letter https://doi.org/10.7554/eLife.64330.sa1
Author response https://doi.org/10.7554/eLife.64330.sa2

## Additional files

### Supplementary files

• Transparent reporting form

### Data availability

All data generated or analysed during this study are included in the manuscript and supporting files. Data for control and COVID-19 bronchoalveolar lavage samples are available in the Sequence Read Archive at NCBI.

The following previously published datasets were used:

| Author(s) | Year | Dataset title | Dataset URL | Database and Identifier |
|---|---|---|---|---|
| Zhou P, Yang XL, Wang XG, Hu B, Zhang L, Zhang W, Si HR, Zhu Y, Li B, Huang CL, Chen HD, Chen J, Luo Y, Guo H, Jiang RD, Liu MQ, Chen Y, Shen XR, Wang X, Zheng XS, Zhao K, Chen QJ, Deng F, Liu LL, Yan B, Zhan FX, Wang YY, Xiao GF, Shi ZL | 2020 | Severe acute respiratory syndrome coronavirus 2 Raw sequence reads | https://www.ncbi.nlm.nih.gov/bioproject/PRJNA605983/ | NCBI BioProject, PRJNA605983 |
| GlaxoSmithKline | 2018 | Microbiome and Inflammatory Interactions in Obese and Severe Asthmatic Adults | https://www.ncbi.nlm.nih.gov/bioproject/PRJNA434133/ | NCBI BioProject, PRJNA434133 |

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
