## [Decision Letter]

**Acceptance summary:**

This manuscript captivated the connection between COVID-19 and cardiovascular disease. Following their previous studies, the authors further investigate the dysregulation of the coagulation and fibrinolytic pathways on COVID-19 patients. The authors' conclusions are well supported, and the authors draw on deep expertise in the field to make exquisite use of gene expression data to obtain deep insights into COVID-19 pathogenesis.

**Decision letter after peer review:**

Thank you for submitting your article "SARS-CoV-2 Suppresses Anticoagulant and Fibrinolytic Gene Expression in the Lung" for consideration by *eLife*. Your article has been reviewed by three peer reviewers, one of whom is a member of our Board of Reviewing Editors, and the evaluation has been overseen by Jos van der Meer as the Senior Editor. The reviewers have opted to remain anonymous.

Summary:

This manuscript extends the authors previous work using transcriptomics data to formulate their "bradykinin storm" hypothesis. Their previous work was extremely high-impact, as it challenged the dogma that a cytokine storm was the principle mechanisms underlying lung pathology in COVID-19. This manuscript further extends their prior work, linking bradykinin signaling to the accumulation of fibrin deposits within pulmonary vessels and the alveolar space, inferred from gene expression data from COVID-19 patients. The authors' conclusions are well supported, and the authors draw on deep expertise in the field to make exquisite use of gene expression data to obtain deep insights into COVID-19 pathogenesis.

Essential Revisions:

1) The reviewer acknowledges the extensive expertise of the impressive author group and congratulates the authors for an interesting investigation. That being said, the reviewer does not think your Abstract reflects the importance or insight of your findings, as your conclusion in the Abstract states "analysis documents increased vascular permeability, enhanced coagulation, and reduced fibrinolytic activity in the lungs of patients with COVID-19". The potential for acute lung injury, hypercoagulability, and microvascular thrombotic sequela of COVID 19 are well known. the reviewer thinks you need to include some of your important aspects of the investigation's findings in the Abstract rather than generalized conclusions.

2) The route of viral entry into the body to the respiratory tract initially triggers the local inflammatory response in the lung, with a host defense mechanism of microvascular thrombotic sequela due to microcirculatory injury, but as the disease progresses without any potential immunity, then multiple other organs may be involved. Again, with the initial insult to the lung and subsequent systemic responses that seem to be the sort of chronology of the infection, as a result, it would be helpful to include some time course of the disease progression that you describe. For example, the skin and other organ involvement is a later, not initial event in most patients. the reviewer thinks your Introduction could be shortened with the extensive literature that exists already about COVID 19 coagulopathy, but also, it would be helpful to provide some sense of the time course of the disease progression. The bronchial alveolar lavage specimens you have our impatience intubated and ventilated most likely, and clearly in an advance disease state. The reviewer thinks that needs to be specifically explained as well.

3) In the subsection “Conclusions”, your first three sentences, including the goal of the present study, need to be incorporated in your Introduction. The reviewer would start your conclusion section with the sentence "We detected pronounced changes in mRNA levels encoding proteins involved in regulation of coagulation, fibrinolysis, and inflammation in the lungs of COVID-19 patients.".

4) Gene and protein naming conventions: when specific proteins or hormones are listed in the text, it would be helpful to also list their formal names, e.g., the reviewer assumes that "tissue factor" in the Abstract refers to human protein CD142 – though later the informal name is given as "extravascular tissue factor (TF)". The informal name, however, is confusing since TF is also the formal symbol for the gene "*Homo sapiens* transferrin (TF)", RefSeq NM_001354704. It would be particularly helpful to list both gene and protein names parenthetically at each such mention. While this is less valuable for those who study the human system directly, it is very important for biologists specializing in model organisms who may be less familiar with medical parlance and informal gene and protein names. It is also extremely useful for AI/ML algorithms that parse Abstracts in search of gene names.

5) Introductory material in the Results section: The Results sections begin with lengthy introductions to the biomedical literature surrounding specific pathways, forcing the reader to scan for novel material. It would be better either to move this material to the Introduction with call-outs to specific Results sections, or, alternatively, to label such introductory paragraphs explicitly so that readers already familiar with the literature can quickly scan past them. As is, it took me a few reads to find the novel result within the first paragraph labeled, "Changes in transcripts encoding proteins in the Kallikrein-Kinin/Bradykinin System".

6) Speculative discussion in the Results section: The final sentence of the first section of Results is speculative, and would fit better in the Discussion:

(6a) "Consistent with our hypothesis that COVID-19 triggers a Bradykinin Storm, the net effect of increased BK production, reduced BK degradation, and increased local expression of BK receptors, may be to enhance vasodilation and vascular permeability that expose blood components to extravascular proteins that promote fibrin formation and contribute to tissue damage."

Or, alternatively, re-word to posit as a conclusion supported by a combination of data in this study and prior literature, perhaps as follows:

"Our observations of increased BK production, reduced BK degradation, and increased local expression of BK receptors, are expected in a system undergoing enhanced vasodilation and vascular permeability, where blood components are exposed to extravascular proteins that promote fibrin formation and contribute to tissue damage. Further, these findings are broadly consistent with our hypothesis that COVID-19 triggers a Bradykinin Storm."

or something like that – at least get rid of the "…may be to…".

(6b) Similarly, we have, "On balance, these data suggest that pulmonary fibrin deposition does not stem from enhanced local TF production and that counterintuitively, COVID-19 may dampen TF-dependent mechanisms in the lungs."

It would be better to use "indicate" than "suggest", or to rephrase something like:

"On balance, these data are inconsistent pulmonary fibrin deposition from enhanced local TF production – but are consistent, if perhaps counterintuitively, with the hypothesis that COVID-19 dampens TF-dependent mechanisms in the lungs."

(6c) The paragraph that begins, "Increased thrombin generation can also be initiated by the plasma protease factor XIIa (FXIIa).(50, 51)…" Appears to be entirely "Discussion" and should probably be removed from Results. Did I miss a new result there?

(6d) The last two sentences of the Results section are also quite speculative:

"The combination of increased fibrinogen expression, reduced production of uPA and uPAR, and a modest increase in PAI-1 suggests a loss of local fibrinolytic capacity. This change may permit the accumulation of fibrin deposits within pulmonary vessels and the alveolar space."

How about the following instead?:

"The combination of increased fibrinogen expression, reduced production of uPA and uPAR, and a modest increase in PAI-1 indicates a loss of local fibrinolytic capacity – consistent with a lung environment permissive of the accumulation of fibrin deposits within pulmonary vessels and the alveolar space (CITE)." and of course a citation should be added where indicated above.

7) Comments on Figures 1 and 2: please increase the resolution of the figure to at least 300dpi and ensure that all fonts are at least size 9. Providing a color scale would make the data more readable for the reader.

8) Table 1: Please provide the log2 data for the observed gene. Adding statistical analysis and providing the statistical significance would make the data more convincing.

9) Materials and methods: In this paper, the authors use 9 COVID-19 patient's BALF samples and 40 control. The reviewer found that in a previous paper (Garvin et al., 2020), the author's group also use the same number of COVID-19 cases and controls. Are these the same samples?

---

## [Author Response]

Essential Revisions:1) The reviewer acknowledges the extensive expertise of the impressive author group and congratulates the authors for an interesting investigation. That being said, the reviewer does not think your Abstract reflects the importance or insight of your findings, as your conclusion in the Abstract states "analysis documents increased vascular permeability, enhanced coagulation, and reduced fibrinolytic activity in the lungs of patients with COVID-19". The potential for acute lung injury, hypercoagulability, and microvascular thrombotic sequela of COVID 19 are well known. the reviewer thinks you need to include some of your important aspects of the investigation's findings in the Abstract rather than generalized conclusions.

We removed general background information from the Abstract and recapitulated the most important results to make the impact of the work clear.

2) The route of viral entry into the body to the respiratory tract initially triggers the local inflammatory response in the lung, with a host defense mechanism of microvascular thrombotic sequela due to microcirculatory injury, but as the disease progresses without any potential immunity, then multiple other organs may be involved. Again, with the initial insult to the lung and subsequent systemic responses that seem to be the sort of chronology of the infection, as a result, it would be helpful to include some time course of the disease progression that you describe. For example, the skin and other organ involvement is a later, not initial event in most patients. the reviewer thinks your Introduction could be shortened with the extensive literature that exists already about COVID 19 coagulopathy, but also, it would be helpful to provide some sense of the time course of the disease progression. The bronchial alveolar lavage specimens you have our impatience intubated and ventilated most likely, and clearly in an advance disease state. The reviewer thinks that needs to be specifically explained as well.

We made major changes to the Introduction. Overall we shortened the Introduction. Some portions of the first paragraph were moved to the "Overview of pathways examined" sections in the Results and Discussion, while some text from the “Conclusions” subsection was incorporated into this section.

We added text relating to disease progression to help put our model into context (e.g., Introduction, second paragraph, second sentence). The Results and Discussion have been changed extensively to make our model more clear.

We added Table 2 with the clinical characteristics of the patients from which the BALF were taken. All were admitted to the ICU, which we note in the new document.

We added text in the Materials and methods section "Study participants" regarding the severity of illness and the condition of the patients at the time of sampling.

3) In the subsection “Conclusions”, your first three sentences, including the goal of the present study, need to be incorporated in your Introduction. The reviewer would start your conclusion section with the sentence "We detected pronounced changes in mRNA levels encoding proteins involved in regulation of coagulation, fibrinolysis, and inflammation in the lungs of COVID-19 patients.".

The first 3 sentences from the subsection “Conclusions” were moved to the Introduction.

4) Gene and protein naming conventions: when specific proteins or hormones are listed in the text, it would be helpful to also list their formal names, e.g., the reviewer assumes that "tissue factor" in the Abstract refers to human protein CD142 – though later the informal name is given as "extravascular tissue factor (TF)". The informal name, however, is confusing since TF is also the formal symbol for the gene "*Homo sapiens* transferrin (TF)", RefSeq NM_001354704. It would be particularly helpful to list both gene and protein names parenthetically at each such mention. While this is less valuable for those who study the human system directly, it is very important for biologists specializing in model organisms who may be less familiar with medical parlance and informal gene and protein names. It is also extremely useful for AI/ML algorithms that parse Abstracts in search of gene names.

We changed gene and protein names throughout the text to use the formal names and include gene and/or protein symbols parenthetically where appropriate.

5) Introductory material in the Results section: The Results sections begin with lengthy introductions to the biomedical literature surrounding specific pathways, forcing the reader to scan for novel material. It would be better either to move this material to the Introduction with call-outs to specific Results sections, or, alternatively, to label such introductory paragraphs explicitly so that readers already familiar with the literature can quickly scan past them. As is, it took me a few reads to find the novel result within the first paragraph labeled, "Changes in transcripts encoding proteins in the Kallikrein-Kinin/Bradykinin System".

We separated these sections into subsections for "Overview of pathways examined" and

"Study findings."

6) Speculative discussion in the Results section: The final sentence of the first section of Results is speculative, and would fit better in the Discussion:(6a) "Consistent with our hypothesis that COVID-19 triggers a Bradykinin Storm, the net effect of increased BK production, reduced BK degradation, and increased local expression of BK receptors, may be to enhance vasodilation and vascular permeability that expose blood components to extravascular proteins that promote fibrin formation and contribute to tissue damage."Or, alternatively, re-word to posit as a conclusion supported by a combination of data in this study and prior literature, perhaps as follows:"Our observations of increased BK production, reduced BK degradation, and increased local expression of BK receptors, are expected in a system undergoing enhanced vasodilation and vascular permeability, where blood components are exposed to extravascular proteins that promote fibrin formation and contribute to tissue damage. Further, these findings are broadly consistent with our hypothesis that COVID-19 triggers a Bradykinin Storm."or something like that – at least get rid of the "…may be to…".

We moved this discussion point to the end of the “Conclusions” subsection.

(6b) Similarly, we have, "On balance, these data suggest that pulmonary fibrin deposition does not stem from enhanced local TF production and that counterintuitively, COVID-19 may dampen TF-dependent mechanisms in the lungs."It would be better to use "indicate" than "suggest", or to rephrase something like:"On balance, these data are inconsistent pulmonary fibrin deposition from enhanced local TF production – but are consistent, if perhaps counterintuitively, with the hypothesis that COVID-19 dampens TF-dependent mechanisms in the lungs."

We changed the text to "indicate" rather than "suggest."

(6c) The paragraph that begins, "Increased thrombin generation can also be initiated by the plasma protease factor XIIa (FXIIa).(50, 51)…" Appears to be entirely "Discussion" and should probably be removed from Results. Did I miss a new result there?

We added text to make the results clear; see text beginning with "Transcripts encoding FXII were slightly decreased (-4-fold)…", which is found in the section "Changes in transcripts encoding intrinsic pathway proteins that initiate thrombin generation" > "Study findings."

(6d) The last two sentences of the Results section are also quite speculative:"The combination of increased fibrinogen expression, reduced production of uPA and uPAR, and a modest increase in PAI-1 suggests a loss of local fibrinolytic capacity. This change may permit the accumulation of fibrin deposits within pulmonary vessels and the alveolar space."How about the following instead?:"The combination of increased fibrinogen expression, reduced production of uPA and uPAR, and a modest increase in PAI-1 indicates a loss of local fibrinolytic capacity – consistent with a lung environment permissive of the accumulation of fibrin deposits within pulmonary vessels and the alveolar space (CITE)." and of course a citation should be added where indicated above.

We changed the text as suggested by the reviewer.

7) Comments on Figures 1 and 2: please increase the resolution of the figure to at least 300dpi and ensure that all fonts are at least size 9. Providing a color scale would make the data more readable for the reader.

Fixed resolution and added color scale.

8) Table 1: Please provide the log2 data for the observed gene. Adding statistical analysis and providing the statistical significance would make the data more convincing.

We added the log2-fold-change values and p-values, as well as the name of the protein for each gene.

9) Materials and methods: In this paper, the authors use 9 COVID-19 patient's BALF samples and 40 control. The reviewer found that in a previous paper (Garvin et al., 2020), the author's group also use the same number of COVID-19 cases and controls. Are these the same samples?

We added a table of the sample IDs (data available from NCBI) to help clarify that these are the same samples. This is listed as Table 1.